# ROCKII inhibition promotes the maturation of human pancreatic beta-like cells

Zaniar Ghazizadeh[1,2], Der-I Kao[1,2], Sadaf Amin[1,2], Brandoch Cook[1], Sahana Rao[1,2], Ting Zhou[1,2], Tuo Zhang [3], Zhaoying Xiang[3], Reyn Kenyon[1,2], Omer Kaymakcalan[1], Chengyang Liu[4], Todd Evans[1] & Shuibing Chen[1,2]

Diabetes is linked to loss of pancreatic beta-cells. Pluripotent stem cells offer a valuable source of human beta-cells for basic studies of their biology and translational applications. However, the signalling pathways that regulate beta-cell development and functional maturation are not fully understood. Here we report a high content chemical screen, revealing that H1152, a ROCK inhibitor, promotes the robust generation of insulin-expressing cells from multiple hPSC lines. The insulin expressing cells obtained after H1152 treatment show increased expression of mature beta cell markers and improved glucose stimulated insulin secretion. Moreover, the H1152-treated beta-like cells show enhanced glucose stimulated insulin secretion and increased capacity to maintain glucose homeostasis after transplantation. Conditional gene knockdown reveals that inhibition of ROCKII promotes the generation and maturation of glucose-responding cells. This study provides a strategy to promote human beta-cell maturation and identifies an unexpected role for the ROCKII pathway in the development and maturation of beta-like cells.

[1] Department of Surgery, Weill Cornell Medical College, New York 10065, USA. [2] Department of Biochemistry, Weill Cornell Medical College, New York 10065, USA. [3] Genomic Resource Core Facility, Weill Cornell Medical College, New York 10065, USA. [4] Department of Surgery, University of Pennsylvania School of Medicine, Philadelphia, Pennsylvania 19104, USA. Zaniar Ghazizadeh and Der-I Kao contributed equally to this work. Correspondence and requests for materials should be addressed to S.C. (email: shc2034@med.cornell.edu)

Human pluripotent stem cells (hPSCs) can potentially provide unlimited starting material to generate functional islets for disease modeling and transplantation therapy of diabetes. Essential to this pursuit is an efficient strategy to differentiate hPSCs into mature pancreatic beta cells. In the past decade, significant progress has been made in directing hPSC differentiation towards this goal. By manipulating signalling pathways known to be involved in pancreatic development, D'Amour et al. showed that hPSCs differentiate into the pancreatic lineage through a stepwise manner[1]. Activation of PKC signalling promotes the generation of pancreatic progenitors[2] and inhibition of the BMP signalling pathway facilitates the generation of insulin-expressing cells[3]. Modifications of the stepwise differentiation approach have been used to generate cells expressing endocrine hormones from both hESCs and hiPSCs[4–10]. Efficient generation of PDX1[+]/NKX6.1[+] pancreatic progenitors facilitates the derivation of single-positive hormonal cells[11, 12]. Most recently, Pagliuca et al.[13] and Rezania et al.[14] reported protocols to generate glucose-responding cells in vitro.

Although these reports have advanced our understanding of beta-cell development and differentiation, the signalling pathways and regulatory network controlling the maturation of human pancreatic beta cells remains incomplete, with only limited progress in this direction. For example, post-translational regulation of the transcription factors ETV1, ETV4, and ETV5 by the ubiquitin ligase COP1 in pancreatic beta cells is critical for insulin secretion[15]. Overexpression of musculoaponeurotic fibrosarcoma oncogene homologue A (MAFA) or thyroid hormone (T3) promotes the functional maturation of human fetal islet-like clusters and hESC-derived beta-like cells[16]. In addition, forced expression of the ligand-dependent transcription factor estrogen-related receptor gamma (ERR gamma) in iPSC-derived beta-like cells enables glucose-responsive secretion of human insulin[17]. Despite these insights, an efficient strategy to promote the maturation of hPSC-derived pancreatic-beta cells is still lacking. Furthermore, although chemicals and growth factors are used in later stages of reported differentiation protocols, the functions of these factors can be unclear, which makes it challenging to optimize the differentiation protocols for each individual hPSC line.

Synthetic small molecules provide useful tools to control stem cell fate, and can also be used to decode the mechanism of biological processes. Previously reported high content chemical screens have identified IDE-1[13] and staurimide[18], which facilitate hPSCs to differentiate toward definitive endoderm. Previously, we identified a PKC activator that promotes the generation of pancreatic progenitors[2], and recently, a small molecule was discovered to facilitate the reprogramming from fibroblasts to the pancreatic lineage[19]. Here, we use a high throughput chemical screen to identify small molecules promoting the maturation of pancreatic beta-like cells from hPSCs and discover a role for ROCKII inhibition in regulation of beta-cell maturation.

## Results

**High content chemical screen and hit validation**. A high content chemical screen was designed to identify small molecules that promote the differentiation from pancreatic progenitors to insulin-expressing (INS[+]) cells. To perform the screen, HUES8 cells were differentiated into a pancreatic progenitor population containing more than 85% PDX1[+] cells using a protocol slightly modified from a previously published version[2] (Fig. 1a). The detailed differentiation protocol is described in Methods. HUES8-derived pancreatic progenitor cells were replated onto

laminin V-coated wells of 384-well plates at 5000 cells/well. After overnight incubation, single compounds from a library containing 4000 chemicals, including FDA-approved drugs, kinase inhibitors, signalling pathway regulators and other annotated compounds, were added to each well at a final concentration of either 10 or 1 μM. After an additional 4 days incubation, cells were stained with an anti-insulin antibody and analyzed for the percentage of INS[+] cells using the Molecular Device ImageXpress[Micro] Automated High Content Analysis System (Fig. 1b). In this assay format, the basal rate of INS[+] cells with dimethyl sulfoxide (DMSO) treatment is $1.0 \pm 0.4\%$. This basal differentiation efficiency of the primary screen is relatively low due to the additional replating step that was needed to decrease well-to-well variation in the high content chemical screen. The compounds that increased the percentage of INS[+] cells by at least five-fold were picked as primary hits, and five hit compounds were identified (Fig. 1c and Supplementary Tables 1 and 2). We chose H1152 (Fig. 1d) for further analysis, since it shows the highest activity with low toxicity (Supplementary Table 1). The immunocytochemistry analysis from the primary screening clearly show both number and percentage of INS[+] cells are higher in a well treated with H1152 than those of a well treated with DMSO control (Fig. 1e). H1152 functions in a dose-dependent manner with $EC_{50}$ at 3.2 μM (Fig. 1f). Because the primary screening strategy required cell replating that decreased differentiation efficiency, we tested H1152 on HUES8-derived pancreatic progenitors without replating the cells. After 8 days treatment, H1152 increases the percentage of INS[+] cells from $12.2 \pm 1.5\%$ to $29.8 \pm 4.1\%$ as shown by intracellular flow cytometry (FCM) (Fig. 1g). Moreover, H1152 treatment also increases the percentage of c-peptide[+] cells indicated by intracellular FCM (Fig. 1g). To determine whether the increase of INS[+] cells depends on the continued presence of H1152, HUES8-derived pancreatic progenitors were treated with 10 μM H1152 for 4 days and switched to H1152-free medium for another 2 days. The percentage of INS[+] cells in wells treated with H1152 is significantly higher than that of the control wells (Fig. 1h), suggesting the increase of INS[+] cells is irreversible and not dependent on the constant presence of H1152. In addition, Ki67 staining shows no difference between DMSO- and H1152-treated cells, indicating that the increase in the percentage of c-peptide[+] cells with H1152 treatment is independent of cell proliferation (Fig. 1i). To evaluate the effect of H1152 on other hESC lines, we tested H1 hESCs, which is relatively efficient for differentiation toward generation of INS[+] cells, and two iPSC lines, which in contrast showed poor differentiation towards generation of INS[+] cells. H1152 treatment increases the percentage of INS[+] cells in each of the cell lines (Fig. 1j, Supplementary Fig. 1a, b), suggesting that the effect of H1152 is cell line independent.

**H1152 promotes the maturation of human beta-like cells**. The primary screen was performed in two dimensional culture to take advantage of image-based quantitative analysis. Considering that islets have a three dimensional structure, we examined the effect of H1152 under such conditions for beta cell generation and maturation. $INS^{w/GFP}$ HES3-derived pancreatic progenitor cells were dissociated with accutase and re-aggregated in three dimensional sphere cultures using low-adherent six-well plates (Fig. 2a). After 8 days culture in 10 μM H1152, the sphere-derived cells were analyzed using flow cytometry based on GFP expression. H1152 treatment significantly increases the percentage and mean fluorescent intensity of INS[+] cells (Fig. 2b). In addition, most of the INS[+] cells co-express NKX6.1 and UCN3, but not glucagon (Fig. 2c). The spheres were further analyzed

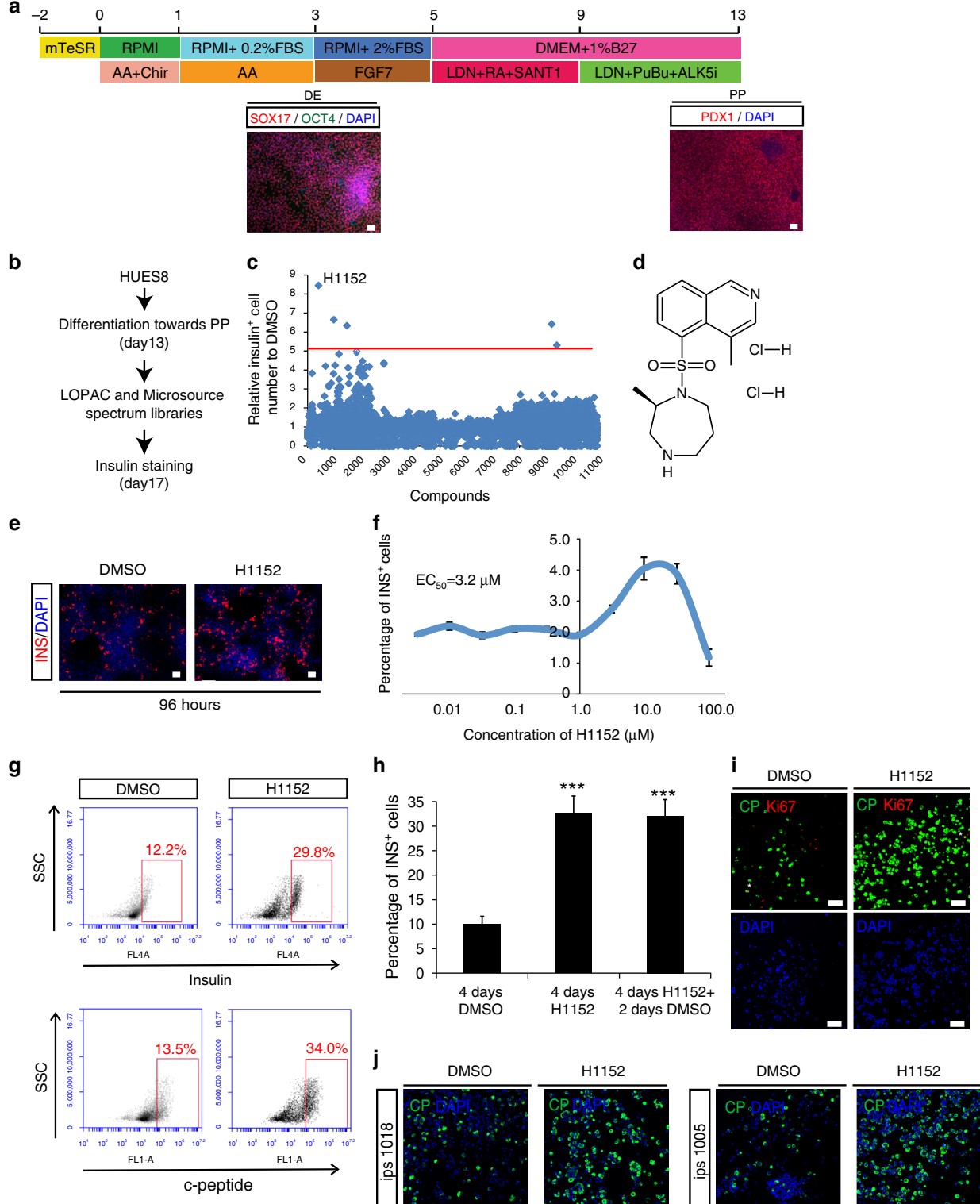

**Fig. 1** High content screening to identify H1152 that promotes the generation of INS+ cells. **a** Scheme of the directed differentiation protocol and representative images of hESC derived definitive endoderm (DE) and pancreatic progenitors (PP). *Scale bar* = 50 μm. **b** Scheme of the chemical screen. **c** Primary screening result. **d** Chemical structure of H1152. **e** Representative images from primary screening. *Red*: insulin/INS, *blue*: DAPI. *Scale bar* = 50 μm. **f** Dose response curve for effects of H1152 on percentage of INS+ cells. **g** Intracellular FCM to detect the expression of insulin (*upper* graphs) and c-peptide (*lower* graphs) of DMSO or H1152-treated cells. **h** The increase of INS+ cells does not depend on the continued presence of H1152. $N = 3$ independent biological replicates. $p$-values were calculated by unpaired two-tailed Student's t-test. ***$p < 0.001$. *Error bar* is SEM. **i** Immunofluorescent imaging of DMSO or H1152 treated cells stained with antibodies against insulin and Ki67. *Scale bar* = 50 μm. **j** Immunofluorescent imaging of DMSO or H1152 treated cells for c-peptide expression in 2 different iPSC lines. *Scale bar* = 50 μm. *AA* Activin A; *RA* Retinoic acid

using intracellular FCM, and H1152 treatment was shown to increase the percentage of NKX6.1$^+$/c-peptide$^+$ cells. The percentage of glucagon$^+$/c-peptide$^+$, somatostatin$^+$/c-peptide$^+$ and pancreatic polypeptide$^+$/c-peptide$^+$ is not significantly changed after H1152 treatment (Fig. 2d and Supplementary

Fig. 2). Results from qRT-PCR experiments using INS-GFP$^+$ cells purified after cell sorting further confirmed the upregulation of pancreatic beta cell markers after H1152 treatment, including transcripts for *NKX6.1, INS, UCN3,* and *G6PC2*. The expression levels of these transcripts in H1152-treated

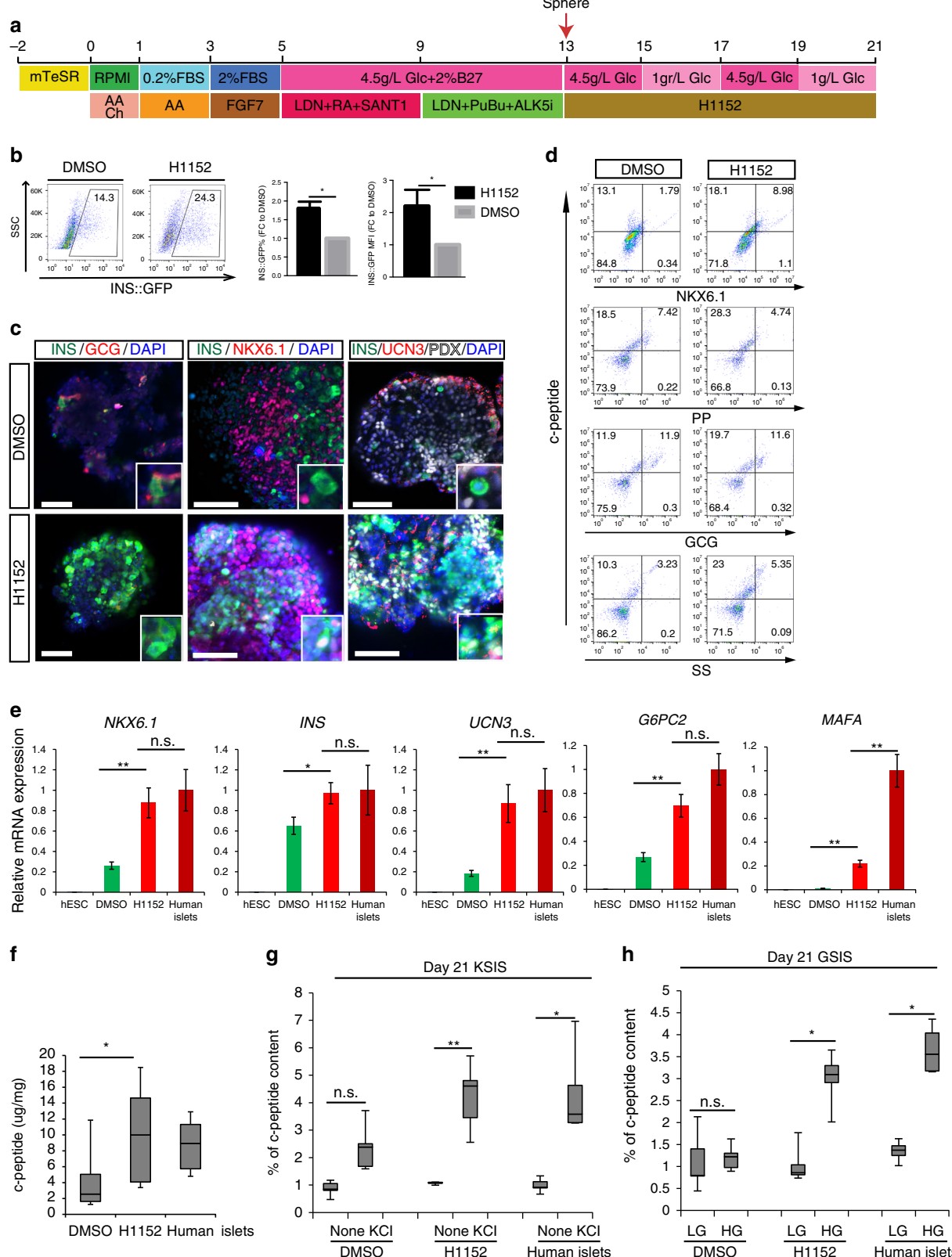

spheres are comparable to levels seen in primary human islets (Fig. 2e). H1152 treatment also significantly increases the expression of *MAFA*. However, we note that the expression level of *MAFA* in INS-GFP$^+$ cells after H1152 treatment is still lower than levels seen in primary human islets (Fig. 2e). Together, the data suggest that H1152 treatment promotes the generation of INS$^+$ cells, and also promotes the expression of mature pancreatic beta cell markers.

The major function of mature pancreatic beta cells is to secrete insulin upon secretagogue and glucose stimulation. To determine the effect of H1152 on cell function, HUES8 cells were differentiated toward pancreatic progenitors and treated with DMSO or H1152. After 8 days treatment, the cells were lysed and analyzed for total c-peptide levels. Primary human islets were used as a positive control. We found that H1152 treatment significantly increases the total c-peptide level of HUES8-derived cells, to levels that are comparable to those in primary human islets (Fig. 2f). To measure the capacity of INS$^+$ cells to respond to secretagogue, H1152-treated or DMSO-treated cells were induced with 30 mM KCl for 30 min. The supernatants were analyzed to measure human c-peptide by ELISA. H1152 treatment increases the levels of secreted c-peptide at both the basal level and after KCl stimulation (Fig. 2g). We starved H1152 treated and control cells for 2 h and then cultured them in 2 or 20 mM glucose. The control cells secrete a similar amount of c-peptide in both 2 and 20 mM glucose conditions, suggesting that they fail to respond to glucose stimulation. In contrast, H1152-treated cells secrete a significantly higher amount of c-peptide than control cells at basal level, and more importantly, secrete an increased amount of insulin upon glucose stimulation (Fig. 2h). The induced level of insulin secretion comparing high and low glucose is comparable to what is seen with human islets.

To comprehensively examine the cellular identity of INS$^+$ cells derived after H1152 treatment, *INS$^{w/GFP}$* HES3 hESCs were differentiated to a pancreatic progenitor population and treated with 10 μM H1152 for 8 days in sphere culture. The INS-GFP$^+$ cells of H1152 or DMSO-treated spheres were purified by cell sorting and transcripts profiled using RNA sequencing (RNA-seq). For comparison, the published data[20] of fetal and adult human primary beta cells were included in the analysis (Fig. 3a). Transcriptional expression analysis showed H1152-treated cells cluster with adult human primary beta cells, while DMSO-treated cells cluster together with fetal human primary beta cells. In the KEGG pathway analysis, the top pathways upregulated in H1152-treated INS-GFP$^+$ cells are insulin secretion and type 2 diabetes mellitus, which include the key genes involved in pancreatic differentiation, including *KCNJ11, ABCC8, GCK, INS, PDX1, SLC2A2, CACNA1A, CACNA1D*, and *GLP1R* (Fig. 3b and c). Higher expression of these genes has been shown to be associated with improved glucose-sensitivity and maturation of pancreatic beta cells[21, 22]. The upregulation of this program supports the notion that H1152 promotes the functional maturation of hESC-derived INS-GFP$^+$ cells. Consistently, genes involved in cell cycle and focal adhesion are downregulated in

H1152-treated cells (Fig. 3b and d), which is in line with previous studies reporting a decline in clonal beta cell expansion in mature islets[17]. Finally, gene set enrichment analysis (GSEA) was used to compare DMSO-treated control cells, H1152 treated pancreatic beta-like cells and, adult and fetal primary human beta cells. The adult beta cell gene set, which includes the genes that are >50-fold higher expressed in adult beta cells compared to levels in fetal beta cells[20], are highly enriched in H1152-treated INS-GFP$^+$ cells; while the fetal beta cell gene set, which includes the genes that are >50-fold higher expressed in fetal beta cells compared to levels in adult beta cells, are highly enriched in DMSO-treated INS-GFP$^+$ cells (Fig. 3e).

**H1152 increases the robustness of optimized protocols.** Recently, two protocols were reported to derive glucose-responding cells from hESCs and iPSCs[13, 23]. We tested whether H1152 promotes the generation and maturation of beta cells on the recently reported Kieffer protocol[14]. Varying doses of H1152 were applied at different stages of the differentiation protocol from *INS$^{w/GFP}$* HES3 (Fig. 4a). We found that 10 μM H1152 shows the strongest effect when added near the completion of the protocol, at Stage 6–2 (day 20–28, Fig. 4a). After 8 days H1152 treatment, the cell clusters were collected and analyzed by confocal imaging, intracellular FCM, qRT-PCR and insulin secretion. Confocal imaging suggested that H1152-treated cell clusters contain more INS$^+$ cells that DMSO-treated cell clusters (Fig. 4b). Consistent with confocal imaging results, intracellular FCM showed that the percentages of NKX6.1$^+$/INS$^+$ cells and glucagon$^-$/INS$^+$ cells are significantly increased following H1152 treatment (Fig. 4c and Supplementary Fig. 3a). In contrast, there was no significant change in the percentage of glucagon$^+$/c-peptide$^+$, somatostatin$^+$/c-peptide$^+$ or pancreatic polypeptide$^+$/c-peptide$^+$ cells with or without H1152 treatment (Fig. 4c). Intracellular FCM analysis on INS-GFP$^+$ cells purified after sorting showed no overlap between glucagon$^+$ and NKX6.1$^+$ populations (Supplementary Fig. 3b). The qRT-PCR results show the upregulation of mature pancreatic beta cell markers, including *NKX6.1, G6PC2, ABCC8, UCN3, NEUROD1, PAX4, KCNK1, KCNK3*, and *MAFA* in the purified INS-GFP$^+$ cells after H1152 treatment (Fig. 4d and Supplementary Fig. 3c). ELISA analysis further validates that H1152-treated cells show higher levels of c-peptide than DMSO-treated cells, without significantly changing the fold induction after KCl treatment (Fig. 4e). Finally, H1152 also promotes the functional maturation of the cells, in that H1152-treated beta-like cell clusters show significantly increased response to 20 mM glucose stimulation (Fig. 4f).

**H1152-treated cells show increased GSIS in vivo.** To determine the functionality of H1152 treated cells in vivo, HUES8 cells were differentiated in three dimensional culture in the presence or absence of 10 μM H1152. At day 28, cell clusters containing around one million cells were collected and transplanted under the left kidney capsule of SCID-beige mice. 3 days after

**Fig. 2** H1152 promotes the maturation of hESC-derived glucose-responding cells. **a** Scheme of the directed differentiation protocol. **b** Flow cytometry analysis, the percentage of INS-GFP$^+$ cells and the mean signal of INS-GFP$^+$ cells of DMSO and H1152 treated spheres. **c–e** Confocal imaging (**c**) intracellular FCM (**d**) and qRT-PCR (**e**) analysis of H1152-treated or DMSO-treated spheres. $N = 3$–6 independent biological replicates. *Scale bar* = 25 μm. *Error bar* is SEM. Primary human islets were used as a control in Fig. 2e. UCN3: urocortin3, SS: somatostatin, PP: pancreatic polypeptide. **f** Total c-peptide content of H1152-treated or DMSO-treated spheres, compared with human islets. **g** KCl-stimulated insulin secretion of H1152-treated or DMSO-treated spheres. **h** GSIS of H1152-treated or DMSO-treated spheres. $N = 8$–12 independent biological replicates. n.s. indicates non-significant difference. *p*-values were calculated by unpaired two-tailed Student's *t*-test. *$p < 0.05$, **$p < 0.01$. *AA* Activin A; *Ch* Chir; *Glc* Glucose; *RA* Retinoic acid; *KSIS* KCl stimulated insulin secretion; *GSIS* Glucose stimulated insulin secretion. The *bottom* and *top* of the box represent the first and third quartiles, the *band* inside the box represents the median. The *ends* of the whiskers represent the minimum and maximum of all the data

transplantation, the mice were treated with 180 mg/kg strepto-zotocin (STZ) to eliminate mouse beta cells. After STZ treatment, all mice become hyperglycemic. The blood glucose level was measured twice per week. 10 days after transplantation, the blood glucose levels start decreasing in mice carrying H1152-treated

cells and in mice carrying DMSO-treated cells. However, the blood glucose level of fed mice carrying H1152 treated cells is significantly lower than for the mice carrying DMSO-treated cells (Fig. 5a). Four weeks post transplantation, the mice were fasted overnight and measured for GSIS (Fig. 5b). Four out of six mice

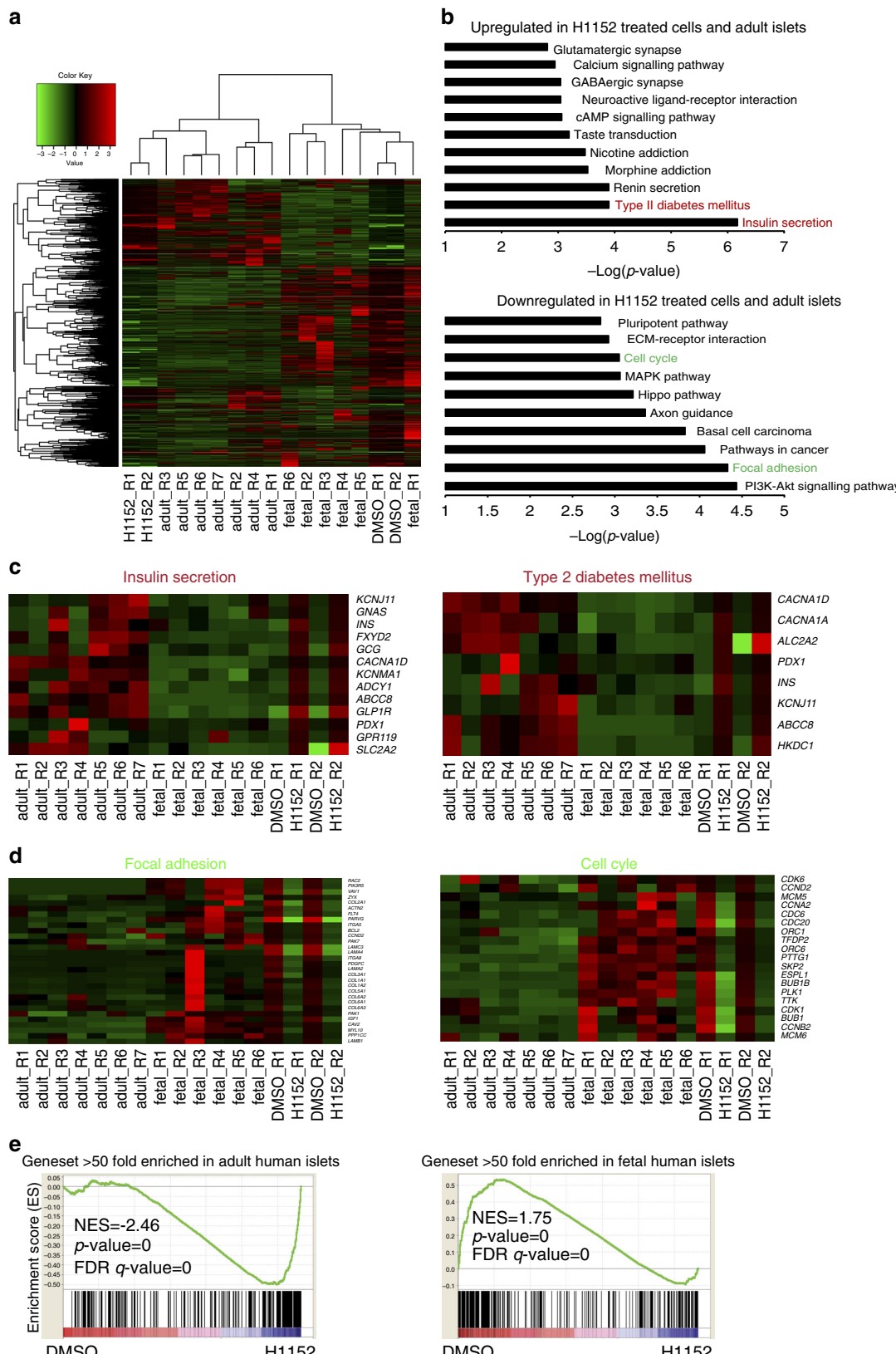

transplanted with H1152-treated cells showed response to glucose stimulation. In contrast, only one of the six mice transplanted with DMSO-treated cells showed response to glucose stimulation. Finally, mice at 5 weeks post transplantation were used to monitor the capacity of the transplanted cells to maintain glucose homeostasis. For Intra-peritoneal glucose tolerance test (IPGTT), the mice were fasted overnight and challenged with 2 g/kg D-glucose by intraperitoneal injection. We found that the mice transplanted with H1152-treated cells show a stronger capacity to maintain glucose homeostasis than DMSO-treated cells (Fig. 5c). The area under the curve for mice carrying H1152-treated cells is significantly lower than for mice carrying DMSO-treated cells (Fig. 5c). After 4.5 months, the kidneys were removed. After kidney removal, the mouse blood glucose level significantly increased and the mice died 1 week later, suggesting that the mice depend on the transplanted human cells to maintain glucose homeostasis (Fig. 5a). Finally, the grafts were analyzed using immunohistochemistry. Higher percentages of INS$^+$, PDX$^+$ and NKX6.1$^+$ cells are detected in the grafts of H1152-treated cells than the grafts of DMSO-treated cells, yet the percentages of glucagon$^+$ and somatostatin$^+$ cells are not significantly different (Fig. 5d and Supplementary Fig. 4). We conclude that H1152-treated cells have improved capacities to respond to glucose stimulation and maintain glucose homeostasis in vivo.

**H1152 functions through ROCKII inhibition**. H1152 was reported to be an inhibitor of Rho-associated kinase (ROCK)[23], which includes both ROCKI and ROCKII. To determine whether H1152 impacts the ROCK pathway in hESC-derived cells, western blotting assays was used to monitor the pathway downstream of ROCK. We found that 48 h treatment of H1152 decreases the phosphorylation of LIMK, the downstream target of ROCK[24] (Supplementary Fig. 5a). To determine whether inhibition of ROCKI or ROCKII facilitates the generation and maturation of human pancreatic beta cells, we generated three $INS^{w/GFP}$ HES3 derivative clonal cell lines, capable of expressing doxycycline (dox)-inducible shRNA against ROCKI, ROCKII or neither (using a control scrambled shRNA sequence). When treated with 10 ng/mL dox, qRT-PCR assays showed that the transcript levels for ROCKI in the $INS^{w/GFP}$ HES3 -shROCKI line-derived cells are decreased by ~ 6.5-fold, and the transcript levels for ROCKII in the of $INS^{w/GFP}$ HES3 -shROCKII line-derived cells are decreased by more than 15-fold, compared to non-dox treated conditions (Supplementary Fig. 5b). Importantly, ROCKII expression levels are not affected by shROCKI and the ROCKI expression levels are not affected by shROCKII, validating the specificity of shRNAs. Western blotting further validated the decrease of ROCKII protein and decrease of LIMK phosphorylation in $INS^{w/GFP}$ HES3-shROCKII treated with dox (Supplementary Fig. 5c). The three cell lines were differentiated to pancreatic progenitors; instead of adding H1152, the cells were treated with 10 ng/ml dox for 8 days during sphere assays (Fig. 6a). The spheres were analyzed with confocal microscopy, intracellular FCM, qRT-PCR, and GSIS. Consistent with H1152-treated spheres, ROCKII-knockdown (KD) spheres, but not

ROCKI-KD spheres, contain a higher percentage of INS$^+$ cells as indicated by intracellular FCM (Fig. 6b). Moreover, the confocal image analysis suggested that only ROCKII-KD spheres, but not ROCKI-KD or scramble-KD spheres, have increased INS$^+$/glucagon$^-$ cells and INS$^+$/NKX6.1$^+$ cells (Fig. 6c). The qRT-PCR analysis on purified INS-GFP$^+$ cells revealed that ROCKII-KD INS-GFP$^+$ cells show increased expression levels of mature pancreatic beta cell markers, including G6PC2, INS, NKX6.1, UCN3 and MAFA (Fig. 6d). Moreover, the ROCKII-KD spheres, but not the ROCKI-KD spheres, show significantly improved responses to KCl or 20 mM glucose stimulation (Fig. 6e and f).

## Discussion

Synthetic small molecules can provide useful tools to control stem cell fate. Previously, we identified small molecules that can function at the early stages of beta cell lineage specification, directing hESC differentiation toward definitive endoderm[25] or pancreatic progenitors[2]. Here we focused on later stages of differentiation and maturation. Although several protocols have been reported to derive glucose-responding cells, the efficiency varies highly among different hESC lines, and this creates a challenge especially for considering eventually deriving patient-specific beta cells. By carrying out a high-content chemical screen, we identified H1152, a ROCK inhibitor, as a compound that promotes the generation and maturation of human pancreatic beta cells in the context of several differentiation conditions. The H1152-treated cells show improved capacities to respond to glucose stimulation and to maintain glucose homeostasis after transplantation into diabetic mice, demonstrating stable maturation of the hESC derived cells. This study provides the field with a useful tool to establish robust protocols for the derivation of mature pancreatic beta-like cells.

The molecular mechanisms controlling the generation and maturation of pancreatic beta cells are largely unknown. The identification of H1152, a known ROCK inhibitor, provides a clue that modulating ROCK signalling is at least part of this mechanism. Indeed, we found that loss of ROCKII, but not ROCKI, is sufficient to induce the generation and maturation of pancreatic beta-like cells. Although not studied in the context of hESC differentiation, previous studies using rat islets and MIN6 cells are consistent with our finding, suggesting that ROCK inhibitors promote glucose-stimulated insulin secretion[26, 27]. ROCK signalling has been suggested to function downstream of glucagon-like peptide 1 to rescue glucotoxicity-induced stress fibers[28]. Another report showed that ROCK and TGFβ1 inhibitors combined with PDX1, NGN3, MAFA, and PAX4 can reprogram human exocrine pancreatic tissue to insulin-producing cells[29]. In addition, ROCK inhibitors have been used to facilitate induction of human beta cell proliferation with WNT-conditioned medium[30]. However, a role for ROCK inhibition in directing human pancreatic development and human pancreatic beta cell maturation has not been previously described. Using both chemical and genetic approaches, we found that inhibition of ROCKII plays an important role in the generation and maturation of pancreatic beta-like cells, independent of promoting GSIS or pancreatic beta

**Fig. 3** Global transcription analysis suggests that H1152 treatment upregulated genes involved in mature pancreatic beta cell function. **a** Heatmap and hierarchical clustering of transcriptional profiling in $INS^{GFP/W}$ HES3-derived INS-GFP$^+$ cells treated with H1152 compared to DMSO treated cells. Fetal and adult primary human beta cells were used as controls (GPL11154). **b** Pathway enrichment analysis on up-regulated or down-regulated genes in $INS^{w/GFP}$ HES3-derived INS-GFP$^+$ cells treated with H1152 and adult human islets. **c** Heatmap of genes in the upregulated pathways, including insulin secretion and type 2 diabetes mellitus. **d** Heatmap of the genes in the downregulated pathways, including focal adherent and cell cycle. **e** GSEA of DMSO-treated, H1152-treated INS-GFP$^+$ cells and fetal and adult human primary beta cells. The dataset are the gene expression level of DMSO-treated or H1152-treated INS-GFP$^+$ cells. The adult beta gene set includes the genes that are >50-fold higher expressed in adult beta cells compared to fetal beta cells. The fetal beta gene set includes the genes that are >50-fold higher expressed in fetal beta cells compared to adult beta cells

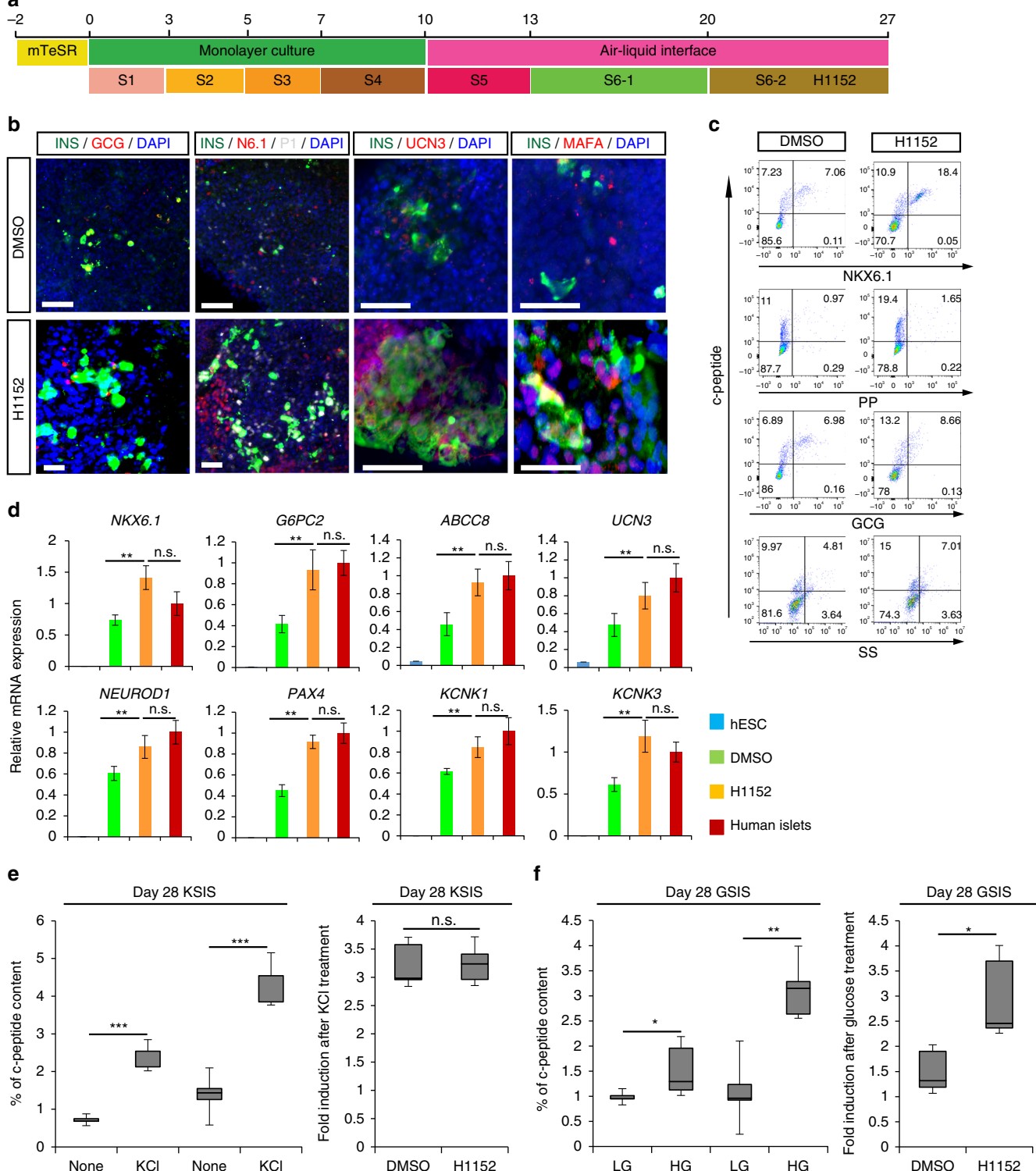

**Fig. 4** H1152 promotes the generation and maturation of hESC-derived glucose-responding cells in the presence of a distinct protocol. **a** Scheme of the directed differentiation. **b**–**d** Confocal imaging (**b**) intracellular FCM (**c**) and qRT-PCR (**d**) analysis of H1152-treated or DMSO-treated cells. *Scale bar*: 100 μm. N = 6–8 independent biological replicates. UCN3: Urocortin3, P1: PDX1, N6.1: NKX6.1, SS: somatostatin, PP: pancreatic polypeptide. *Error bar* is SEM. **e** The percentage of c-peptide content and fold induction of H1152-treated or DMSO-treated cells after exposure to 30 mM KCl. Fold induction was calculated by dividing the percentage of c-peptide content of KCl treated condition by that of the untreated condition. N = 8–12 independent biological replicates. **f** The percentage of c-peptide content and fold induction of H1152-treated or DMSO-treated cells in the presence of 2 mM D-glucose (LG) and 20 mM D-glucose (HG). Fold induction was calculated by dividing the percentage of c-peptide content of HG treated condition by that of the LG treated condition. N = 8–12 independent biological replicates. *p*-values were calculated by unpaired two-tailed Student's t-test. *$p < 0.05$, **$p < 0.01$. *KSIS* KCl stimulated insulin secretion, *GSIS* Glucose stimulated insulin secretion. The *bottom* and *top* of the box represent the first and third quartiles, the *band* inside the box represents the median. The *ends* of the whiskers represent the minimum and maximum of all the data

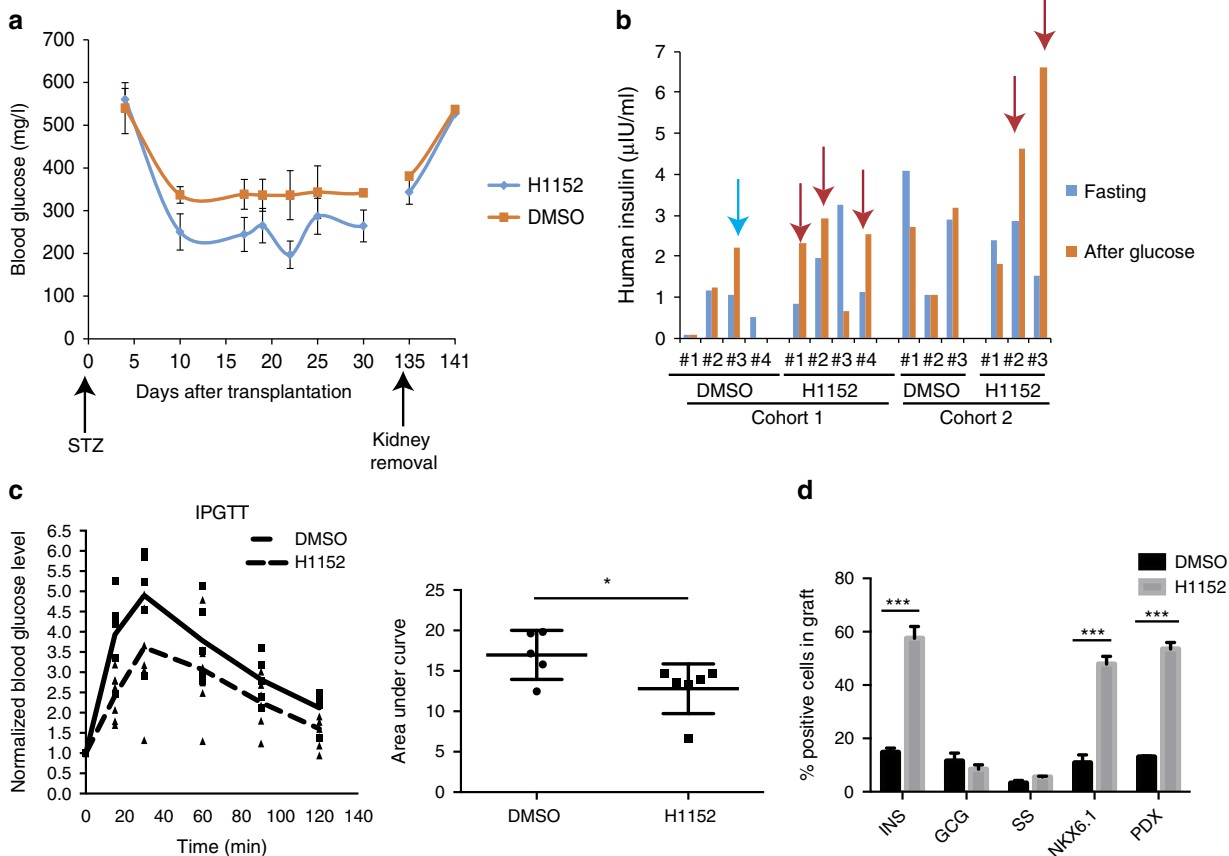

**Fig. 5** H1152-treated cells show improved capacity to respond to glucose stimulation and maintain glucose homeostasis in vivo. **a** Blood glucose of mice transplanted with H1152-treated or DMSO-treated cells. The cells were transplanted at day −4. 180 mg/kg STZ was given at day 0. The kidneys were removed at day 134. *Error bar* is SEM. **b** GSIS of mice 5 weeks after transplantation with H1152-treated or DMSO-treated cells. The mice were fasted overnight and injected with 3 g/kg D-glucose. *Red* and *blue* arrows indicate positive GSIS in mice transplanted with H1152- or DMSO-treated cells, respectively. **c** IPGTT of mice 5 weeks after transplanted with H1152-treated or DMSO-treated cells. The mice were fasted overnight and injected with 2 g/kg D-glucose. The blood glucose level was measured every 15 min. $N = 6$ mice for each condition. Fold induction was calculated by dividing the blood glucose level at each time point by the baseline glucose level. *Error bar* is SEM. **d** Quantification of immunohistochemistry analysis of grafts of H1152-treated cells or DMSO-treated cells $N = 3$ independent biological replicates. *Error bar* is SEM. *STZ* Streptozotocin. *p*-values in **a** were calculated by two-way repeated measures ANOVA with a Bonferroni test. *p*-values in **c** were calculated by unpaired two-tailed Student's *t*-test. *$p < 0.05$, **$p < 0.01$

cell proliferation. This provides a new tool for hESC/iPSC differentiation, and highlights a key molecular mechanism controlling generation and maturation of human pancreatic beta cells. Finally, ROCK pathway activity might potentially be useful as an indicator for hESC/iPSC differentiation capacities.

## Methods

**hESC culture and differentiation**. The H1 line was purchased from WiCell Institute. HUES8 and 1005 iPSCs were kindly provided by Harvard Stem Cell Institute. The INS$^{GFP/W}$ HES3 line was kindly provided by the laboratory of Dr. Edward Stanley at the Monash University. The 1018 iPSCs was a gift from Dr. Dieter Egli at Columbia University.

HUES8 used in the primary screening were routinely cultured on irradiated MEF feeders in Dulbecco's modified Eagle's medium (DMEM)/F12 supplemented with 20% KnockOut Serum Replacement, 2 mM Gluta-MAX, 1 mM nonessential amino acids, 1.1 mM β-merceptoethanol, 10 ng/ml basic fibroblast growth factor, and 50 ng/ml normacin.

HUES8, H1, or INS$^{GFP/W}$ HES3 hESCs used in the experiments from Figure 2 to Figure 6 were grown on Matrigel-coated 6-well plates in mTeSR1 medium (STEM CELL Technologies).

H1152 was purchased from Tocris. Human adult islets were provided by the NIH-funded Integrated Islet Distribution Program. The de-identified donors ranged in age from 40, 48, 53, 58, 59, or 63 years old.

**High Content Chemical Screen**. To perform the high content chemical screen, HUES8 cells were cultured on feeders to 80–90% confluency, then treated with 3 μM CHIR99021 (CHIR, Stem-RD) and 100 ng/ml Activin A (R&D systems) in

RPMI (Cellgro) supplemented with 2 mM GlutaMAX and 100 U/ml Pen/Strep for 1 day, and then 100 ng/ml Activin A in RPMI supplemented with 0.2% fetal bovine serum (FBS), 2 mM GlutaMAX and 100 U/ml Pen/Strep. The medium was changed 2 days later to 50 ng/ml FGF7 (PeproTech) in RPMI supplemented with 2 mM GlutaMAX, 100 U/ml Pen/Strep and 2% FBS, and maintained for an additional 2 days. Cells were transferred to 300 nM LDN193189 (LDN, Axon), 2 μM retinoic acid (Sigma), and 0.25 μM SANT-1 (Sigma) for the first 4 days and then to 300 nM LDN193189 (LDN, Axon), 20 nM Phorbol 12,13- dibutyrate (PuBu, Sigma), and 1 μM ALK5i (Enzo) in DMEM supplemented with 2 mM GlutaMAX, 100 U/ml Pen/Strep, and 1x B27 (Invitrogen) for an additional 4 days.

To perform the screening, HUES8 cell-derived pancreatic progenitors were plated on laminin V-coated 384 well plates at 5000 cells/80 μl medium/well (DMEM supplemented with 1X B27). After overnight incubation, cells were treated with compounds from a chemical collection containing the LOPAC library and Microsource Spectrum libraries, with one compound per well at either 10 and 1 μM. DMSO treatment was used as a negative control. Detailed information of primary screening is listed in Supplementary Table 1. After 96 h incubation, cells were fixed and stained using an insulin antibody (DAKO). Plates were analyzed using a Molecular Devices ImageXpress High-Content Analysis System (Molecular Devices). Compounds increasing the percentage of INS$^+$ cells by more than 5-fold were picked as primary hits.

**Stepwise differentiation**. HUES8, H1, or INS$^{GFP/W}$ HES3 hESCs cells were differentiated into beta-like cells using a strategy adapted from a recent protocol[14]. Briefly, hESCs were plated on Matrigel (BD Biosciences, 354234)-coated dishes at $10^5$ cells per cm² in mTeSR medium. Differentiation was initiated 48 h after plating when the culture was about 80% confluent. The differentiation protocol is summarized in Supplementary Table 3.

**Insulin secretion assay**. About 10–15 spheres at day 21 or 4–5 clusters of S6 cells at day 28 ($5 \times 10^5$ cells) were rinsed twice with Krebs buffer (129 mM NaCl, 4.8 mM KCL, 1.2 mM $MgSO_2$, 1 mM $Na_2HPO_4$, 1.2 mM $KH_2PO_4$, 2.5 mM $CaCl_2$, 5 mM $NaHCO_3$, 10 mM HEPES, 0.1% BSA, in deionized water) starved in 1 ml glucose-free DMEM (Gibco, 97060-876; with 2 mM L-glutamine) for 1 h in a 5% $CO_2$/37 °C incubator. After washing with warmed KRBH, cells were incubated in KRBH (with 0.1% BSA) for 1 h in an air incubator at 37 °C. Cells were then incubated in 1 ml KRBH with combinations of 2 mM glucose, 20 mM glucose, 30 mM KCl, in an air incubator at 37 °C for 1 h. 200 μl medium were collected from each well and filtered by a 0.45 μm filter for ELISA analysis (Human C-peptide ELISA kit, Millipore, EZHCP-20K). Cells were lysed to measure the protein levels in each sample for normalization to total c-peptide content (Human C-peptide

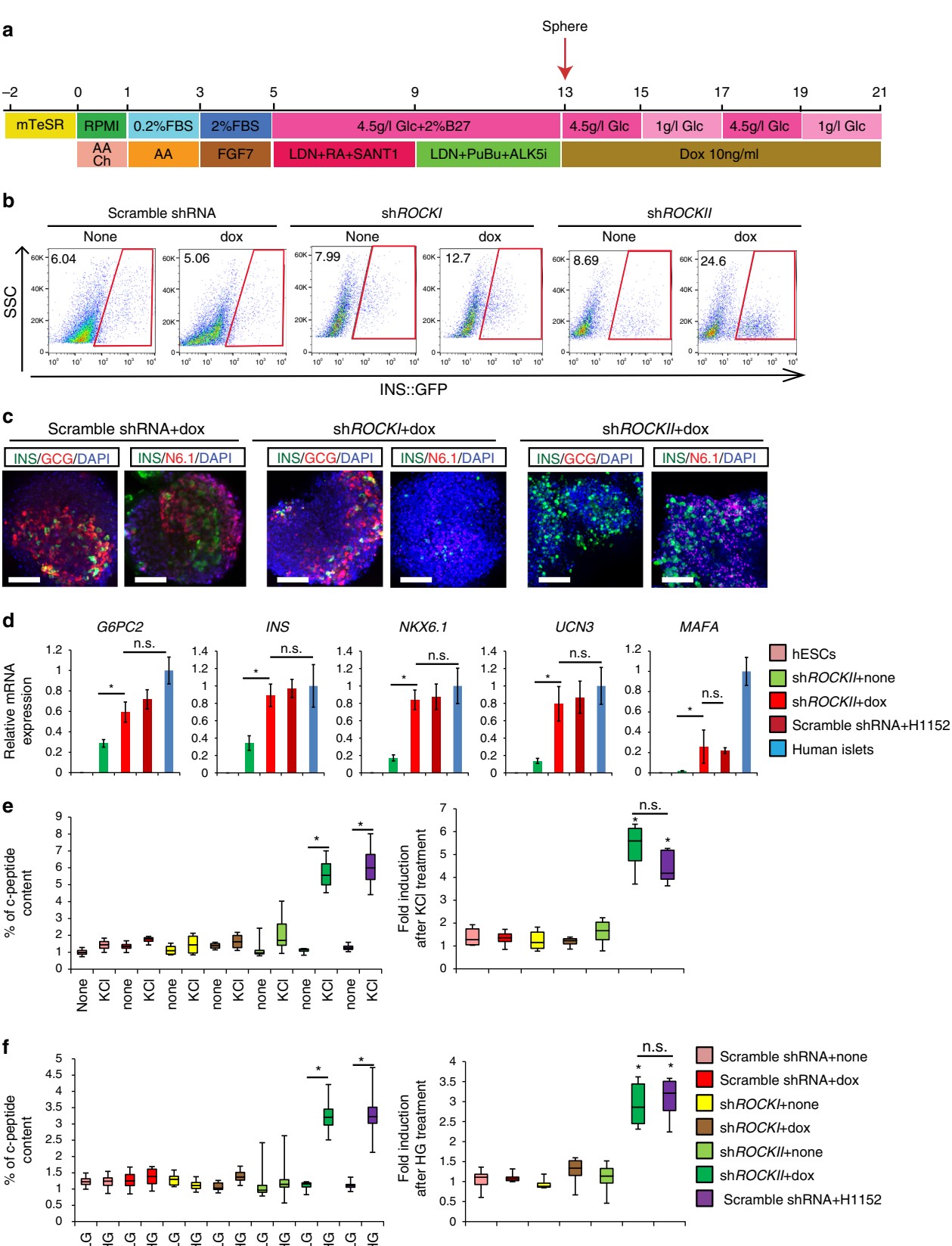

ELISA kit, Millipore, EZHCP-20K). 30–50 human islets isolated from six different donors were used for GSIS analysis as control.

**In vivo Transplantation GSIS and IPGTT.** All mouse procedures were performed following NIH guidelines, and the protocols were approved by the Weill Cornell Medical College Institutional Animal Care and Use Committee (IACUC). We used 6- to 8-week-old male SCID-Beige mice for these studies. Animals were randomly selected for various treatment models (H1152 or DMSO treated cells). Animals were anesthetized with ketamine/xylazine (90–120 mg/kg ketamine/body weight; 10 mg/kg xylazine/body weight) under aseptic conditions. Cell clusters containing around $2 \times 10^6$ cells were transplanted under the kidney capsule. 5 days after transplantation, 180 mg/kg streptozotocin was injected intraperitoneally to destroy mouse pancreatic beta cells. Blood glucose levels were monitored twice a week with a glucometer (freestyle lite).

To perform GSIS, mice were starved for about 20 h before the test. Mouse blood was collected under fasting conditions and at 30 min after intraperitoneal injection with 3 g/kg glucose solution. The mouse sera at fasting condition and after glucose stimulation were analyzed using the ultrasensitive human insulin ELISA kit (ALPCO, 80-INSHUU-E01.1). To perform IPGTT analysis, the mice were fasted overnight and treated with 2 g/kg glucose. Blood glucose level (mg/dl) in each animal was measured before and every 15 min in the first hour and every 30 min in the second hour after glucose injection.

**RNA sequencing.** For RNA sequencing, total RNA was extracted using the Agilent RNA nanoprep kit (Agilent Technologies, 400753). The cDNA libraries were generated using TruSeq RNA Sample Preparation (Illumina). Each library was sequenced using single-reads in HiSeq2000/1000 (Illumina). Gene expression levels were analyzed using Cufflinks[31, 32].

**Bioinformatics analysis.** For heat map normalization, RPKM values were normalized per gene over all samples. Specifically, mean and standard deviation (stdev) of RPKM were calculated for each gene, and used to linearly transform RPKM using the formula (rpkm−mean)/s.d. The normalization was done separately for DMSO vs. H1152-treated cells and fetal vs. adult human beta cells due to different methods for RNA-seq sample preparation. The heat map was then generated by heatmap.2 in the R gplots package. The distance function used in the heatmap clustering is calculated based on Pearson correlation of expression values of selected genes between samples. Pathway analysis on up/down-regulated genes in H1152-treated group was performed using DAVID function annotation tool (http://david.abcc.ncifcrf.gov/). The p-value was used to determine the significance of enrichment or overrepresentation of terms for each annotation.

**GSEA analysis.** The GSEA analysis was performed using GSEA software (Broad Institute). The data set are the gene expression level of DMSO-treated or H1152-treated INS-GFP$^+$ cells purified from cell sorting. The adult beta gene set includes the genes that are >50-fold higher expressed in adult beta cells compared to levels in fetal beta cells. The fetal beta gene set includes the genes that are >50-fold higher expressed in fetal beta cells compared to levels in adult beta cells[20].

**Immunofluorescence analysis.** For immunofluorescence (IF), cells were fixed with 10% (v/v) formalin for 20 min at room temperature (RT) and then blocked and permeabilized using 0.3% (v/v) PBS/triton X-100 (Sigma) and 5% heat-inactivated horse serum for 1 h at room temperature and followed by incubation with primary antibody at 4C overnight. Primary antibodies were anti-SOX17 (1:500, R&D), anti-OCT4 (1:400, Santa-Cruz Biotechnology), anti-Glucagon (1:2000, Sigma), anti-PDX1 (1:500, R&D), anti-NKX6.1 (1:500, University of Iowa Hybridoma bank), anti-UCN3 (1:500, Atlas Antibodies), anti-MAFA (1:500, Bethyl Laboratories) and anti-insulin (1:500, DAKO). After washing three times, cells were incubated with secondary antibodies for 1 h at room temperature. Alexa Fluor secondary antibodies were obtained from Invitrogen (1:500). The images were quantified using MetaMorph Image Analysis Software (Molecular Device).

For cryosectioning, the grafts were isolated and then fixed using 4% PFA (eBiosciences) at 4 °C for 4 h. After an additional 2 days incubation in 30% sucrose (Fisher Scientific) solution at 4 °C, the fixed grafts were embedded in OCT (Fisher Scientific) and sections with 5 µm thickness was prepared. The sections were blocked with 5% heat-inactivated horse serum in 0.3% (v/v) PBS/triton X-100 solution. The sections were then stained with appropriate primary antibodies overnight at 4 °C. Secondary antibodies were added for 1 h at room temperature. The stained sections were then incubated with 1 ng/ml DAPI (Sigma) and washed several times before they were mounted with Fluoro-Gel (Electron microscopy Sciences). The images were analyzed using inverted fluorescents microscope (Olympus) or confocal microscope (LSM 880).

**Intracelluar FCM.** hESC-derived cells were dissociated by Accutase (STEM CELL Technologies). The resulting single cell suspension was fixed and permeabilized using FOXP3 staining buffer set (eBiosciences) for 1 h at room temperature following manufacturer's instruction. The cells were then stained with primary antibodies at 4C overnight. Primary antibodies were anti-NKX6.1 (1:500), anti-Glucagon (1:500, cell signalling), anti-c-peptide (1:500, University of Iowa Hybridoma bank), anti-insulin (1:500), anti-somatostatin (1:500, Novus biologicals) and anti-pancreatic polypeptide (1:1000, DAKO). After washing, cells were incubated with secondary antibodies for 1 h at room temperature. Alexa 647 conjugated donkey anti-mouse or Alexa 488 conjugated donkey anti-guinea pig secondary antibody (1:500) Flow cytometry data were obtained using an Accuri C6 flow cytometry instrument and analyzed with FlowJo.

**Quantitative RT-PCR analysis.** Total RNA was isolated using Qiagen RNeasy mini kit according to instructions by the manufacturer. 0.5 µg of total RNA was used to generate first strand cDNA using the Superscript III First Strand Synthesis System (Thermofisher). SYBR Green-based qPCR was carried out using the Roche 480 Lightcycler. Triplicate reactions were carried out for each sample. GAPDH was used as a control to normalize target gene expression. Sequences of primers used are listed in Supplementary Table 4.

**Cloning and generation of dox-inducible shRNA in hESC lines.** The pLKO-Tet-On vector[33] was utilized to create shRNA expressing constructs. Briefly, the stuffer DNA was removed from PLKO-Tet-On by AgeI/EcoRI digestion and replaced with double-stranded oligos encoding the desired shRNA and AgeI/EcoRI sites. shRNA sequences were derived from previously published experiments[34] and are listed in Supplementary Table 5.

To generate lentivirus for infection, 293T cells were transfected with a mixture containing 4 µg of shRNA-encoding plasmid and 2 µg of each of gag-pol, rev and env plasmids in 200 µL of Xtremegene transfection reagent (Roche). The supernatant were collected at 48 and 72 h intervals and concentrated using Lenti-X concentrator (Clontech). HUES8 cells were infected with lentiviruses in the presence of 8 µg/ml polybrene. 3 days after infection, cells were selected with 5 mg/mL hygromycin (Gibco) for 3 days and subcloned. Single colonies were selected and expanded for further confirmation.

**Western blotting analysis.** HUES8-derived cells were harvested at the times indicated directly into complete ice-cold lysis buffer (20 mM Tris (pH 7.5), 150 mM NaCl, 50 mM NaF, 1% NP-40 substitute, and Thermo Scientific HALT protease inhibitor cocktail 1:100). Lysates were loaded onto 10% NuPage Bis-Tris precast gels (Invitrogen), resolved by electrophoresis, and proteins were transferred to polyvinylidene fluoride (PVDF) membranes (Bio-Rad). Membranes were blocked with 5% BSA in TBS + 0.05% Tween and incubated in primary antibody overnight.. The antibodies were anti-rabbit phospho-LIMK1/2 (1:250 dilution) and total LIMK1 (1:500 dilution; both from Cell Signalling), anti-mouse RockII (1:2000 dilution; Santa Cruz Biotechnology), and anti-mouse beta-actin (Sigma, 1:50,000 dilution). Membranes were washed and incubated with horseradish peroxidase-conjugated secondary antibodies (BioRad) for 1 h in 5% milk-TBS-0.05% Tween, and developed using SuperSignal West Pico ECL substrate (Thermo Fisher Scientific). All uncropped western blots can be found in Supplementary Fig. 6.

---

**Fig. 6** ROCKII inhibition promotes the generation and maturation of glucose-responding cells. **a** Scheme of the directed differentiation protocol. **b–d** Intracellular FCM (**b**), Confocal imaging (**c**) and qRT-PCR analysis (**d**) of $INS^{w/GFP}$ HES3-carrying vectors to express shROCKI, shROCKII or scrambled shRNA in the presence or absence of 10 ng/ml dox treatment. Primary human islets and H1152-treated cells were used as controls in **d**. Error bar is SEM. **e** The percentage of c-peptide content and fold induction of ROCKI-KD and ROCKII-KD cells after 30 mM KCl treatment. Fold induction was calculated by dividing the percentage of c-peptide content in the KCl treated condition by that of the untreated condition. $N = 8$–12 independent biological replicates. **f** The percentage of c-peptide content and fold induction of ROCKI-KD and ROCKII-KD cells in the presence of 2 mM D-glucose (LG) and 20 mM D-glucose (HG). Fold induction was calculated by dividing the percentage of c-peptide content in the HG treated condition by that of the LG treated condition. Scale bar: 25 µm. Primary human islets and H1152-treated cells were used as controls in **d**. H1152-treated cells were used as controls in **e** and **f**. $N = 8$–12 independent biological replicates. p-values were calculated by unpaired two-tailed Student's t-test. *$p < 0.05$, **$p < 0.01$. N6.1, NKX6.1. The bottom and top of the box represent the first and third quartiles, the band inside the box represents the median. The ends of the whiskers represent the minimum and maximum of all the data

**Statistical analysis**. Data are presented as mean SEM and were derived from at least three independent biological replicates if not otherwise specifically indicated. n.s. indicates non-significant difference. Statistical analysis was performed using unpaired two-tailed Student's t-test if not otherwise specifically indicated. For multiple comparisons between DMSO or H1152 treated cells p-values were calculated by one-way repeated measures ANOVA or two-way repeated measures ANOVA with a Bonferroni test. $*p < 0.05$, $**p < 0.01$, $***p < 0.001$, and $****p < 0.0001$.

**Data availability**. The authors declare that all data supporting the findings of this study are available within the article and its supplementary information files or from the corresponding author upon reasonable request. RNA-seq data have been deposited in the NCBI Gene Expression Omnibus database under accession code GSE84325.

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

## Acknowledgements

This work was supported by The New York Stem Cell Foundation (R-103, S.C.), NIDDK (DP2 DK098093-01, DP3DK111907-01), and American Diabetes Association (1-17-IBS-019, 1-12-JF-06), a Tri-Institutional Starr Stem Cell Postdoctoral Fellowship (D.K.). S.C. is New York Stem Cell Foundation-Robertson Investigator. This study was also supported by a Shared Facility contract to T.E. and S.C. from the New York State Department of Health (NYSTEM C029156). Mouse anti-NKX6.1, anti-NKX2.2 and rat anti-c-peptide antibodies were purchased from the Developmental Studies Hybridoma Bank at the University of Iowa. We are also very grateful for technical support and advice provided by Harold S. Ralph in the Cell Screening Core Facility, and the Flow Cytometry Core Facility at the Hospital for Special Surgery, NY.

## Author contributions

Conceptualization, S.C.; Methodology, D.K., Z.G, B.C., T.Z., X.Z., S.R., O.K., and S.A.; Manuscript preparation: D.K., Z.G., T.E. and S.C.; Funding Acquisition, S.C. and T.E.; Resources, C.L.

## Additional information

**Competing interests:** The authors declare no competing financial interests.

