## [Peer Review File · Nature Communications]

This manuscript has been previously reviewed at another journal that is not operating a transparent peer review scheme. This document only contains reviewer comments and rebuttal letters for versions considered at Nature Communications.

REVIEWERS' COMMENTS:

Reviewer #1 (Remarks to the Author):

The authors have done an outstanding job responding to the various comments. Their work has made this manuscript much stronger as a result. Very pleased to recommend acceptance.

--

Reviewer #2 (Remarks to the Author):

The authors did a good job in addressing previous concerns. However, there are still several points unclear in the text, which should be clarified.

1. There are several new cell lines used in the revised MS, of which the origins are not provided, including INS GFP/W HES3 hESCs, and Ips 1018, Ips 1006. The authors should clarify the access to the cell lines, their origins, and whether the authors established the cell lines themselves, if so then the establishment should be referred in the text.

Response: We have added in the Methods section the description and source of hPSC lines including INS GFP/W HES3 hESCs, Ips 1018, and Ips 1006.

2. Flow cytometry (FCM) should be used instead of FACS, since FACS is only one of the methods used in FCM.

Response: We have changed all “FACS” to “FCM” in the main text.

3. Human islets: adult and fetal pancreas should be defined. The age, and embryonic days should be provided.

Response: The human islets were isolated from adult pancreas. The age information for the islets has been added in the Method section.